# Transcription Factor CEBPB Inhibits the Expression of the Human *HTR1A* by Binding to 5′ Regulatory Region In Vitro

**DOI:** 10.3390/genes10100802

**Published:** 2019-10-12

**Authors:** Yong-Ping Liu, Xue Wu, Jing-Hua Meng, Mei Ding, Feng-Ling Xu, Jing-Jing Zhang, Jun Yao, Bao-Jie Wang

**Affiliations:** School of Forensic Medicine, China Medical University, Shenyang 110122, China; liuyongping1993@163.com (Y.-P.L.); wuxue19910425@163.com (X.W.); jhmeng@cmu.edu.cn (J.-H.M.); dingmei77@126.com (M.D.); xufengling1992@163.com (F.-L.X.); fayixiaojing@163.com (J.-J.Z.); yaojun198717@163.com (J.Y.)

**Keywords:** *HTR1A* CCAA/T enhancer binding protein beta (CEBPB), transcription regulation, 5-HT1A receptor, mental diseases

## Abstract

This study identified a transcription factor that might bind to the 5′ regulatory region of the *HTR1A* and explored the potential effect on 5-HT1A receptor expression. Based on JASPAR predictions, the binding of the transcription factor was demonstrated using the electrophoretic mobility shift assay (EMSA). Vectors over-expressing the transcription factor were co-transfected into HEK-293 and SK-N-SH cells with the recombinant pGL3 vector, and relative fluorescence intensity was measured to determine regulatory activity. Additionally, the qRT-PCR and Western blot were also used to identify whether the transcription factor modulated the endogenous expression of 5-HT1A receptor. The results suggest that the transcription factor CCAA/T enhancer binding protein beta (CEBPB) likely binds to the −1219 to −1209 bp (ATG+1) region of the *HTR1A*. Two sequences located in the −722 to −372 bp and −119 to +99 bp were also identified. Although the effect of CEBPB on endogenous 5-HT1A receptor expression was not significant, it exhibited the strong inhibition on the relative fluorescence intensity and the mRNA level of *HTR1A*. CEBPB inhibited the human *HTR1A* expression by binding to the sequence −1219–−1209 bp. This is useful and informative for ascertaining the regulation of 5-HT1A receptor and mental diseases.

## 1. Introduction

The serotonin system not only maintains normal regulation of the immune system [1], but also influences mood [2,3], emotion [4] and numerous neuropsychiatric disorders [5,6,7,8,9,10,11,12]. The related experiments in vivo and vitro have confirmed that these neuropsychiatric diseases might have changes in the serotonin neurotransmitters expression levels [13,14] or metabolic abnormalities, which eventually caused the excessive inhibition or activation of the downstream pathways [15,16]. 

As a neurotransmitter, serotonin exerts its function by binding to the corresponding serotonin receptor [17]. Of the 14 serotonin receptor subtypes [18], the 5-HT1A receptor has attracted a lot of attentions in specific fields. The presynaptic 5-HT1A receptor (auto-receptor) is located in the dorsal raphe nucleus of the brain and plays a considerable role in controlling the tone of the 5-HT system [19,20,21]. The expression of the 5-HT1A auto-receptor increased in patients with depression and schizophrenia, which resulted in the decreasing of the neurotransmitter level [22]. The 5-HT1A receptor also acts as a postsynaptic receptor to regulate the physiological functions of serotonin in the hippocampus, amygdala and hypothalamus [23,24]. However, whether the abnormal expression of this receptor contributes to the mental illness is still controversial [25,26]. It is becoming increasingly necessary to understand the effects of the regulatory region in the *HTR1A* on receptor expression, which may further clarify the pathogenesis of psychiatric diseases.

Because the *HTR1A* only has one exon and no intron, it appears that the 5′- regulatory region is critical to 5-HT1A receptor expression. A study of the human *HTR1A* 5′- regulatory region indicated that the basic transcription sequence was located in the upstream of the gene at approximately 715 bp (ATG+1), which was also identified by a series of positive regulatory elements, such as PET-1, SP1, NFκB and MAZ I/II/III/IV [8,27]. In the −1250 bp to −1200 bp region of the *HTR1A*, the presence of a negative regulatory element was considered likely to be the glucocorticoid response element nGRE [28]. High expression of the response element in the hippocampus renders the 5-HT1A receptor in this region more sensitive to glucocorticoid and stress [29]. Double inhibitory regions in the rat and human *HTR1A* at −1620 bp to −1517 bp were bound by Freud-1 and REST transcription factors, respectively [30]. Both Freud-1 and REST independently regulated *HTR1A* expression in cells that express 5-HT1A [31]. 

By constructing the recombinant vectors containing different length sequences of the 5′- regulatory region, we explored several regions that exhibited regulatory effects on the gene expression. Among the sequences, the relative fluorescence intensity of the fragment between −1409 bp and −1124 bp was significantly different in two cell lines [32]. Therefore, the bioinformatics analysis and related experiments were conducted to verify the possible transcription factor binding to the target sequence. The effects of the transcription factor on the messenger RNA (mRNA) and protein expression of *HTR1A* were also further clarified.

## 2. Materials and Methods 

### 2.1. Bioinformatics Analysis and Prediction of Transcription Factors

A previous study showed that the *HTR1A* fragment −1409 bp to −1124 bp had strong transcriptional regulatory activity. JASPAR software (http://jaspar.genereg.net/cgi-bin/jaspar_db.pl) was used to predict the possible transcription factors that identified the target sequence with a perfect match >80%. As a supplement, the relevant literature and the PubMed database (https://www.ncbi.nlm.nih.gov/pubmed/) were searched to screen the transcription factors that were expressed in both cell lines or involved in the nervous, dopamine, or serotonin systems.

### 2.2. Electrophoretic Mobility Shift Assay (EMSA)

Nucleoproteins from HEK-293 and SK-N-SH cell lines were obtained using the nucleus and cytoplasmic protein extraction kit with 1% protease inhibitor phenylmethanesulfonyl fluoride (PMSF) (Beyotime, Shanghai, China). Protein quantification was determined using the bicinchoninic acid (BCA)method (Beyotime). The synthetic 5′-end biotin-labeled probe was consistent with the transcription factor binding sequence. The 5′-end biotin-labeled, unlabeled specific competitive probe and the non-specific competitive mutation probe were generated by Taihe Biotechnology Co. (Beijing, China). The relevant sequences of the probes were showed in Table 1. In each 20 μL reaction system, 20 fmol labeled probe were incubated with 8–9 μg nucleoprotein, 4 pmol specific competitive probe or non-specific mutation probe in an ionized environment, including 2 μL 10×binding buffer, 1 μL 50% glycerol, 1 μL 100 mM MgCl_2_, 1 μL 1 μg/μL Poly(dI.dC) and 1 μL 1% NP-40 for 20 min at room temperature. The only difference in the supershift system is that the reaction also contained 0.4–0.6 μg primary antibody for each transcription factor. After pre-electrophoresis of the 6% non-denaturing polypropylene gel at 100 V for 1 h, a 20 μL reaction system containing 5 μL 5×EMSA loading buffer was loaded and electrophoresed for 50–60 min. Transfer was at 100 V for 75 min with the positive charge nylon membrane (Millipore, MA, USA), followed by 20 min cross-linking at a 254 nm wavelength with UV light. Each cross-linked nylon membrane (8 × 5 cm) was soaked in 5 mL blocking buffer and gently shaken for 20 min. The stabilized streptavidin-HRP conjugate was then mixed with blocking buffer at a ratio of 1:300, and the nylon membrane was incubated in the same manner. Protein-DNA migration bands were detected with the Tanon-5500 chemiluminescence imaging analysis system (Tanon Science & Technology, Shanghai, China) with a mixture of stable peroxide solution and luminol/enhancer solution (Thermo Scientific™, LightShift™ Chemluminescent EMSA kit, Waltham, MA, USA).

### 2.3. Construction of pGL3 Recombinant Vector

The pGL3 recombinant vector containing the sequence −1208 bp to +99 bp was constructed by Taihe Biotechnology Co. (Beijing, China). The target fragment within the *HTR1A* at −1227 bp to +99 bp was amplified by PCR using the forward primer: 5′-GAAGATCTCTCTCCCGGTTCCCCAAC-3′ and the reverse primer: 5′-CCCAAGCTTGTCGGAGATACCAGTAGTGTT-3′. *Bgl*II and *Hind*III restriction endonucleases were introduced into the 5′-end. The purified target gene was cloned into a PGM-T vector and then sub-cloned into the pGL3 vector. The recombinant vector was successfully screened by sequencing to prepare for eukaryotic cell transfection.

### 2.4. Cell Culture

Human embryonic kidney HEK-293 cells were cultured in HyClone^®^ DMEM high glucose medium containing 10% fetal bovine serum (Thermo Fisher Scientific, Waltham MA, USA), 5% CO_2_ + 95% mixed air at 37 °C. Neuroblastoma SK-N-SH cells were treated with KeyGEN BioTECH^®^ DMEM high glucose medium with 0.110 g/L sodium pyruvate containing 15% fetal bovine serum under the same conditions.

### 2.5. Transient Transfection of pGL3-HTR1A-1235, pGL3-HTR1A-1227, pGL3-HTR1A-1208 and pGL3-HTR1A-1196 Recombinant Vectors

Cells with a density of 90% were inoculated into 24-well plates at 2 × 10^5^ cells per well and cultured for 36–48 h. The pGL3 recombinant vectors were then co-transfected with the Renilla luciferase expression vector pRL-TK (Promega, Madison, WI, USA) using Lipofectamine^®^ 2000 reagent (Invitrogen, Carlabad CA, USA). Luciferase and renin luciferase expression was detected after 24 h.

### 2.6. Transient Transfection of pGL3-HTR1A Recombinant Vectors with CEBPB Over-Expressing Vector

When cells were in an exponential growth phase, 2–4 μL of Lipofectamine^®^ 2000 reagent was incubated with 500 ng pGL3-HTR1A recombinant vectors, 500 ng pEGFP-N1-CEBPB or pEGFP-N1-Basic over-expressing vector and 50 ng pRL-TK control vector. After 24 h of co-transfection, green fluorescent protein (GFP = 33KD) was observed under fluorescence microscopy, and cell lysates were collected.

### 2.7. Transfection of CEBPB Over-Expressing Vector

When cell density reached 90% or more, 1 × 10^6^ cells per well were inoculated into 6-well plates. A total of 3500 ng pEGFP-N1-CEBPB or pEGFP-N1-Basic over-expressing vector was transfected into HEK-293 and SK-N-SH cell lines using Lipofectamine^®^ 3000 reagent. Total protein was collected after 48 h of transfection.

### 2.8. Dual Luciferase Reporter Assay

Cells in each well were lysed with 100 μL 1×PLB. A total of 20 μL luciferase substrate and 1×STOP reagent were added to 30 μL cell lysates to detect luciferase (LUC) and Renilla luciferase (TK) protein expression (Promega). The relative fluorescence intensity was the ratio of firefly luciferase expressionand renilla luciferase expression, therefore LUC/TK.Each sample was tested in triplicate per experiment with a total of three experiments.

### 2.9. Quantitative Real-Time PCR

Total RNA from HEK-293 or SK-N-SH+pEGFP-N1-Basic and HEK-293 or SK-N-SH+pEGFP-N1-CEBPB cells was extracted according to the following procedures. 800 μL Trizol was added to each 6-well plates, and the cell lysate was successively treated with chloroform, isopropanol and 75% ethanol. Total RNA was then dissolved in 30 μL diethyl pyrocarbonate (DEPC)water for several hours and quantified using a UV spectrophotometer. The complementary DNA (cDNA) reaction system included 5 × PrimeScript Buffer 4 μL, PrimeScript RT Enzyme Mix I 1 μL or PrimeScript RT Enzyme Mix I-free (negative control), Oligo dT Primer 1 μL (50 μmol/L), Random 6 mers 1 μL (100 μmol/L), Total RNA (1000 ng) and RNase free dH_2_O (Takara, Dalian, China). The reaction conditions were 37 °C for 15 min, 85 °C for 5 s, and 4 °C. The real time PCR reaction system included SYBR Green Premix Ex Taq II 10 μL, forward primer and reverse primer 1.6 μL (5 pmol/ μL), ddH_2_0 4.8 μL and cDNA 2 μL. Reaction conditions: pre-denaturation: 95 °C 30 s; denaturation: 95 °C 5 s, 60 °C 30 s, 40 cycles; dissolution curve: 95 °C 15 s, 60 °C 30 s, 95 °C, 15 s.

### 2.10. Western Blot

Total protein from HEK-293, HEK-293+pEGFP-N1-Basic and HEK-293+pEGFP-N1-CEBPB cells was extracted by NP-40 and PMSF. Protein from SK-N-SH cells was collected as described. Quantified protein samples were separated by electrophoresis in 10% denatured polypropylene gel and transferred using polyvinylidene fluoride (PVDF) membrane for 1 h at 100 V. After blocking with 8% skimmed milk, primary antibodies for 5-HT1A (Thermo Scientific™) and βactin (Abbkine, CA, USA) were diluted with TBS-T at a ratio of 1:1000 and 1:2000, respectively. Membranes were incubated overnight at 4 °C. Diluted secondary antibody (Abbkine) (1:5000) was incubated with the membrane, which had been washed three times with TBS-T for 2 h. Target protein expression was detected by Lumino (ECL)luminescent solution and the Tanon-5500 chemiluminescence imaging analysis system. Each sample was tested in duplicate per experiment, and three separate experiments were conducted.

### 2.11. Statistical Analysis

The relative fluorescence intensity is expressed by LUC/TK. Nine LUC/TK values were obtained for each sample and expressed as the mean ± standard deviation (SD). One-way analysis of variance (ANOVA) was used to analyze differences between multiple samples, while the least significant difference (LSD) *t*-test was performed to compare two samples. Real-time PCR was calculated by the 2^−ΔΔCT^ method to compare differences in mRNA expression. Endogenous 5-HT1A receptor expression was normalized with β-actin, and the difference in the grayscale values was determined by the Student’s *t*-test. *p* < 0.05 indicated a statistically significant difference.

## 3. Results

### 3.1. Prediction and Screening of Transcription Factors

Using JASPAR software, the transcription factors FEV, CEBPB and LMX1B recognized the −1235 bp to −1196 bp fragment, successively. According to the PUBMED database, CEBPB functions as a zinc finger structure transcription factor and affects the expression of many genes in the nervous system [33]. This study focused on the potential role of this transcription factor in the regulation of the *HTR1A* (Figure 1). 

### 3.2. Sequence Location Identified by CEBPB 

Electrophoretic Mobility Shift Assay was performed to detect the binding of the transcription factors to target sequences using nuclear protein extracted from HEK-293 and SK-N-SH cells. The DNA-protein complex was observed in the reaction using a *HTR1A* probe (−1235–−1196) in the 5′ regulatory region of the *HTR1A*. At least three proteins were combined with this fragment (Figure 2a, Figure 2b). Within the three different length fragments between −1235 bp and −1196 bp that might bind to proteins, we found that the transcription factor CEBPB recognized the sequence located at −1219 bp to −1209 bp. Subsequently, the supershift experiments further clarified that CEBPB was the regulatory element at this position in the HEK-293 cells (Figure 2c). 

### 3.3. Significant Differences in Relative Fluorescence Intensity of pGL3-HTR1A-1235, pGL3-HTR1A-1227, pGL3-HTR1A-1208 and pGL3-HTR1A-1196 Recombinant Vectors

The results of one-way analysis of variance showed that the regulatory activities of the four recombinant vectors were significant differences with the *p* = 1.42 × 10^−6^ and *p* = 2.42 × 10^−6^ in the two cell lines. We found that the relative fluorescence intensities of adjacent target fragments was statistically significant in HEK-293 cell lines (*p* = 0.015, 0.003 and *p* = 7.08 × 10^−7^, respectively). Similar expression trends were observed in SK-N-SH cells with the statistically significant differences of *p* = 0.02, *p* = 0.009 and *p* = 6.53 × 10^−7^, respectively. Importantly, the relative fluorescence intensity increased significantly when the −1219 bp–−1209 bp sequence recognized by CEBPB was truncated, suggesting that CEBPB might inhibit the gene expression (Figure 3).

### 3.4. CEBPB Significantly Inhibited HTR1A Expression

When the recombinant vectors pGL3-HTR1A-1227, pGL3-HTR1A-1208, pGL3-HTR1A-1196 were co-transfected with the pEGFP-N1-CEBPB or pEGFP-N1-Basic, the relative fluorescence intensities were detected. One-way analysis of variance showed that the relative fluorescence intensities of all target fragments were significantly statistically different (*p* < 0.05). In HEK-293 cells, the relative fluorescence intensity was significantly inhibited by the CEBPB with *p* = 6.88 × 10^−18^, 3.11 × 10^−18^ and 2.90 × 10^−18^. Similar down-regulation of the gene expression was observed in SK-N-SH cells (*p* = 1.28 × 10^−13^, 8.02 × 10^−10^ and 1.77 × 10^−13^). Additionally, the rates of decline were 64.81% ± 0.046, 58.55% ± 0.045 and 65.29% ± 0.041 and the decline degree of adjacent sequences showed significantly different in HEK-293 cells with *p* = 0.006 and 0.004, respectively. With the down-regulation degree of 64.39% ± 0.061, 55.70% ± 0.034 and 67.10% ± 0.058 in SK-N-SH cells, the inhibitory effect of CEBPB on the −1227 bp, −1208 bp and −1196 bp sequences was significant difference (*p* = 0.002 and 0.0001) (Figure 4).

In the two cell lines, all recombinant vectors, including pGL3-HTR1A-1124, pGL3-HTR1A-1064, pGL3-HTR1A-908, pGL3-HTR1A-722, pGL3-HTR1A-119 and pGL3-HTR1A-372, had responses to CEBPB and relative fluorescence intensities reached statistical significance. Since all recombinant vectors containing the *HTR1A* shared a common sequence −722 bp–−372 bp and exhibited the significant responses to CEBPB, the −722 bp–−372 bp fragment might be an important potential recognition region for CEBPB. The results also showed that the inhibition degree of CEBPB for the fragments −1124 bp–+99 bp and −1124 bp–−119 bp decreased from 65% to 20%. This reduction in the inhibition indicated that the sequence −119 bp–+99 bp might be bound by CEBPB (Figure 5).

### 3.5. The Effect of CEBPB on the Endogenous 5-HT1A Receptor Expression

Although, western blot showed that the CEBPB transcription factor had no significant effect on endogenous 5-HT1A receptor expression in two cell lines (*p* > 0.05) (Figure 6), the mRNA expression level of *HTR1A* was significantly inhibited by the CEBPB in HEK-293 and SK-N-SH cells with *p* = 0.045 and 0.027, respectively (Figure 7).

## 4. Discussion

We found that the sequence 5′-GGTTCCCCAAC-3′ located at −1219 bp to −1209 bp in the *HTR1A* might be the target fragment identified by the transcription factor CEBPB. Moreover, CEBPB had a similar inhibitory activity in pGL3-HTR1A-722 and pGL3-HTR1A-372. Thus, the sequence ranging from −722 bp to −372 bp was another potential DNA binding domain for the transcription factor. The results also reflected that even if the sequence −1219 bp–−1209 bp was truncated, all recombinant vectors containing the common fragment −722 bp–−372 bp revealed the strong response to the CEBPB. Studies have emphasized that approximately 750 bp upstream of the *HTR1A* promoter region is essential for the basal transcription and has numerous transcription factors binding. Therefore, we suspected that this area might be more critical to the identification and function of CEBPB. Moreover, the inhibitory role of CEBPB on the −1124 to +99 and −1124 to −119 fragments was significant and descending rates were approximately 65% and 20%, respectively. Decreasing relative activity of the luciferase gene was possibly due to the deletion of the fragment (−119 bp to +99 bp), which might be bound by CEBPB. The dual luciferase reporter assay showed that CEBPB exerted a significant inhibitory effect on *HTR1A* expression. The inhibitory effect of CEBPB appeared to be contrary to some studies regarding the role of CEBPB in increasing the gene expression [34]. However, CEBPB had three protein isoforms: LAP* (38KD), LAP (35KD) and LIP (20KD) [35]. Subtype LIP that exhibits inhibitory effects could be produced by the LAP* and LAP subtypes [36]. Moreover, studies on the transcription factor Deaf-1 have shown that it exhibits an up-regulation function on the expression of 5-HT1A auto-receptor and an inhibitory effect on the 5-HT1A post-synaptic receptor expression [14]. The two receptors encoded by the *HTR1A* showed differences in the distribution regions. This indicated that the regulation of the same transcription factor might be altered by the different genes or tissues. Therefore, the inhibitory effect of CEBPB on the *HTR1A* could be identified. Related study have demonstrated that lipopolysaccharide-induced depression-like rat was accompanied by an increase in the CEBPB (38KD) expression [37]. The substantial evidences supported the reduction in *HTR1A* or 5-HT1A receptor expression in patients with depression and schizophrenia. Therefore, we solidly demonstrated that a significantly increased CEBPB that recognizes the *HTR1A* strongly inhibited the level of this gene and leaded to the susceptibility to the mental illness. CEBPB is also involved in the neuroinflammation. The inflammatory disorders have also been found in the patients with schizophrenia. Thus, the important role of the transcription factor CEBPB in regulating *HTR1A* expression might be a new target for the treatment of mental diseases. 

To fully demonstrate the inhibitory effect of CEBPB on gene expression, pGL3-HTR1A-1227 and pGL3-HTR1A-1208 recombinant vectors were constructed. The relative fluorescence intensities of the adjacent sequence were significantly different in both of the cell lines. The results suggested that the regulatory elements that had an inhibitory effect on transcriptional activity might be combined with −1227 bp to −1208 bp, which was completely consistent with the regulation of CEBPB at −1219 bp to −1209 bp. 

Because 5-HT1A receptor expression was demonstrated in both cell lines, the effect of CEBPB on endogenous receptor expression was examined using qRT-PCR and Western blot. Although the endogenous protein expression of 5-HT1A receptor did not significantly change at 24 h, 48 h, 72 h, 84 h and 96 h after the two cell lines were transfected with the transcription factor (data not fully shown), its mRNA expression was significantly inhibited by CEBPB (*p* < 0.05). These results indicated that CEBPB also played a significant down-regulation role to the endogenous *HTR1A*. Insignificant inhibition of protein levels was inconsistent with results of the dual luciferase reporter assay. This is likely due to several reasons. First, the sensitivity of Western blot is lower than the dual luciferase reporter experiment [33]. Thus, minor changes induced by CEBPB in the endogenous receptor may not be easily detected. Second, CEBPB is essential for regulation of the immune system [38,39] and is also involved in neuropsychiatric disorders [37] and nervous system-related signaling pathways [40]. In the brain striatum, dopamine neurotransmitter responses were adjusted by CEBPB, followed by variations in cAMP and PKA-related signaling [33]. The serotonin system is not an independent system that interacts with the dopamine and gamma-aminobutyric acid systems [41]. Therefore, we hypothesized that the increased expression of the CEBPB in cells might lead to the activations of other regulatory factors, neurotransmitters or signaling pathways. These factors may exhibit the synergistic effects, but are more likely to perform the opposite functions of the CEBPB in order to maintain the cellular homeostasis. Eventually, these factors that exert the different roles might make the inhibition of the endogenous 5-HT1A receptor expressions by the CEBPB weakened and not obvious [42]. However, the CEBPB was not considered to have no effect on the expression of the endogenous 5-HT1A receptor. 

Although this study suggests a new transcription factor that may regulate *HTR1A* expression, there are several limitations. EMSA confirmed that the CEBPB protein bound to the *HTR1A* in vitro. However, the inhibition of the endogenous 5-HT1A receptor expression by the CEBPB was not significantly observed. We hypothesized that the low cell transfection efficiency of the CEBPB over-expression vector might result in insufficient inhibition of 5-HT1A receptor expression. The role of transcription factors also requires other co-factors. In terms of the unsystematic conditions of the individual cell lines, in vivo experiment may yield more accurate and convincing results.

## 5. Conclusions

This study identified CEBPB, as a transcription factor significantly inhibited *HTR1A* expression and its primary binding site was located in the sequence −1219 bp to −1209 bp (ATG+1). In addition, −722 bp to −372 bp in the 5′ regulatory area of the *HTR1A* and the −119 bp to +99 bp in the 5′-UTR region may also be DNA-binding domains of CEBPB. Although CEBPB did not show obvious inhibitory effects on endogenous 5-HT1A receptors, the *HTR1A* mRNA expression was significantly down-regulated by CEBPB. It was still required for 5-HT1A expression given that the transcription factor was involved in regulation of the nervous system. Further research will be necessary to elucidate mechanisms involved in regulating the *HTR1A*. 

## Figures and Tables

**Figure 1 genes-10-00802-f001:**
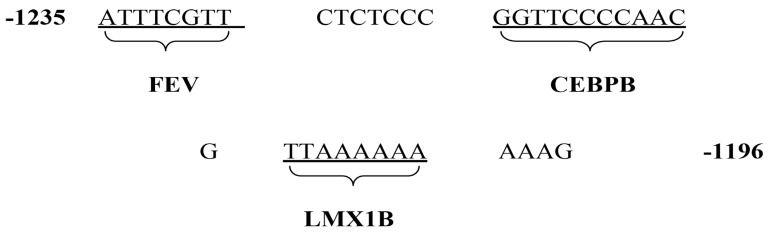
Prediction of transcription factor in the human *HTR1A* based on JASPAR software and related literature, the effect of the transcription factor CEBPB on the *HTR1A* expression were evaluated. FEV and LMX1B transcription factors may target fragments adjacent to the CEBPB binding sequence. Numbers represent sequence location (ATG+1), and bases recognized by the transcription factors are underlined.

**Figure 2 genes-10-00802-f002:**
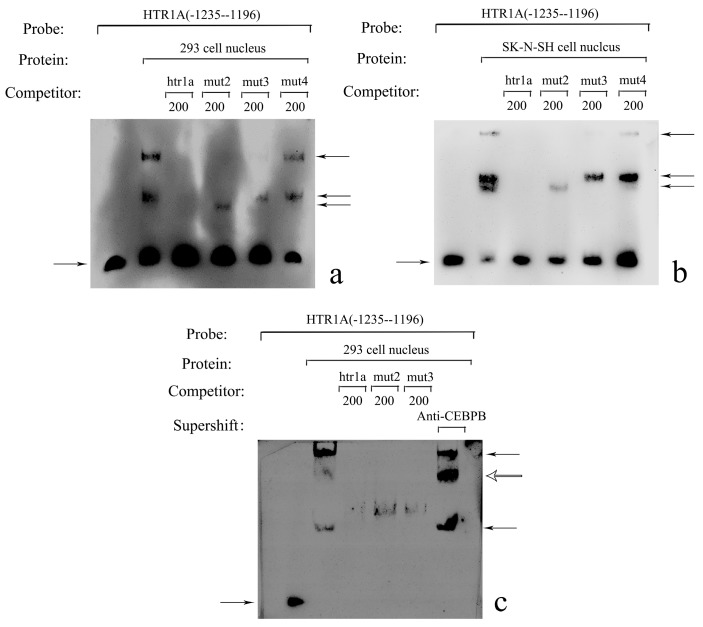
(**a**) EMSA identified CEBPB bound to the −1219 bp to −1209 bp fragment, (**b**) A synthesized 5′-end biotin-labeled probe was located at −1235 bp to −1196 bp. The unlabeled specific competitive sequence was consistent with the labeled probe, whereas the non-specific competitive sequences mut2, mut3 and mut4 were mutated according to the fragments identified by CEBPB, FEV and LMX1B, respectively. All lanes contained the probe. In addition to the first lane on the left, nuclear protein extracted from HEK-293 or SK-N-SH cells was loaded. The third to the sixth lanes showed probe mixed with the specific or non-specific competitive sequences. Protein hysteresis suggested that at least three proteins combined with this sequence in both cell lines. The fragment recognized by CEBPB might be located at −1219 bp to −1209 bp. (**c**) In the supershift assay, the components of the first to fifth lanes were identical to that of the Figure 2a,b. In particular, the sixth lane compared with the second lane additionally contained the CEBPB-specific antibody. A more delayed protein-DNA binding band was observed in HEK-293 cells. This suggests CEBPB was bound to −1219 bp to −1209 bp in the HEK-293 cells. The bands indicated by the left arrows were the probes and the right arrows were the protein-DNA complexes. The hollow arrow pointed to the supershift band.

**Figure 3 genes-10-00802-f003:**
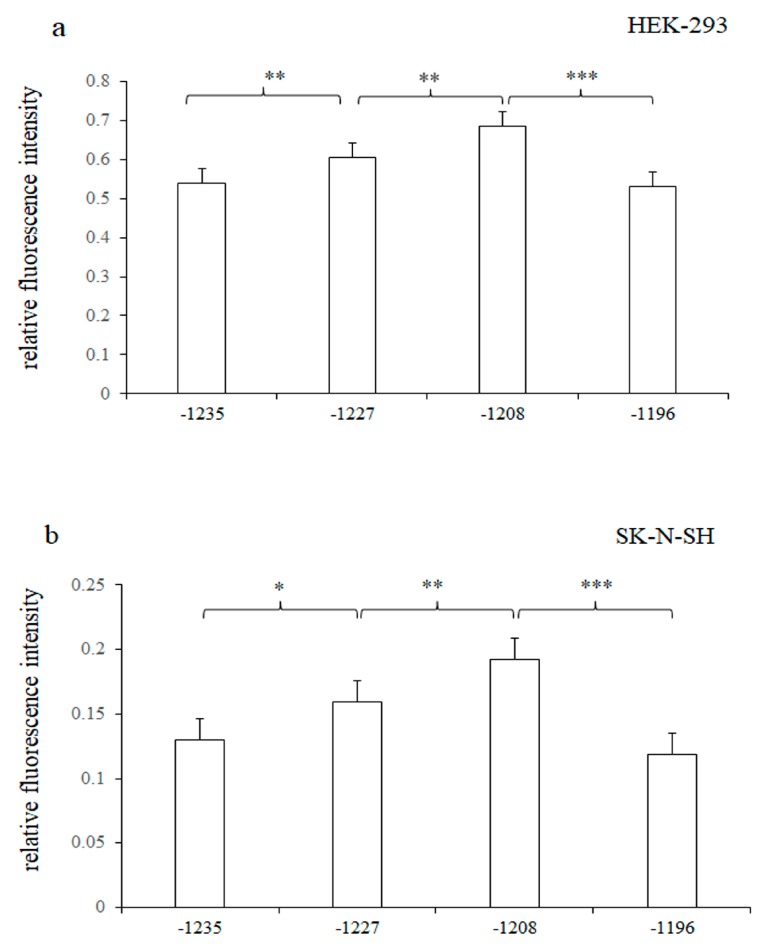
(**a**) Relative fluorescence intensity of pGL3-HTR1A-1235, pGL3-HTR1A-1227, pGL3-HTR1A-1208 and pGL3-HTR1A-1196, (**b**) In HEK-293 and SK-N-SH cells, the relative fluorescence intensity of the adjacent sequence showed a statistically significant difference. Relative fluorescence intensity of each sample was normalized as the mean ± standard deviation (SD), and the least significant difference (LSD) *t*-test was used to compare the two samples. * 0.02 < *p* < 0.05, ** 0.00 < *p* <= 0.02, *** *p* <= 0.001.

**Figure 4 genes-10-00802-f004:**
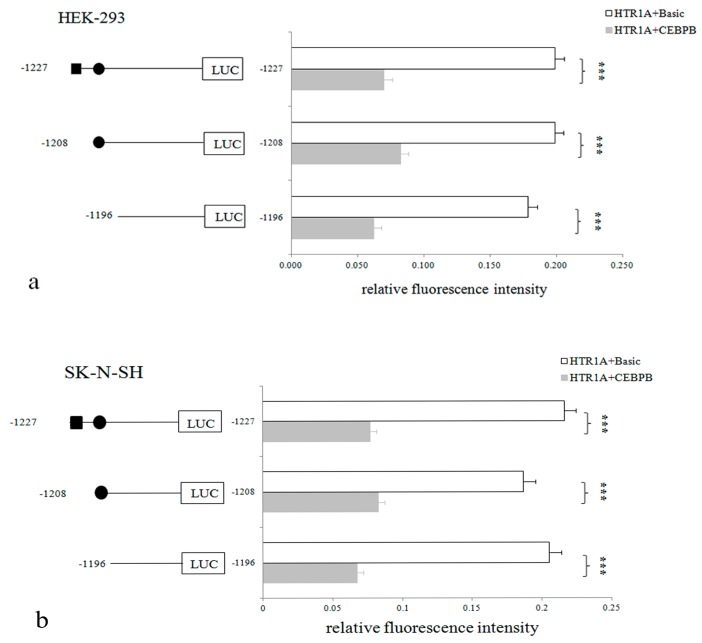
(**a**) Effect of CEBPB on the relative fluorescence intensity in two cell lines, (**b**) The rectangle and circle represent the sequence positions identified by CEBPB, LMX1B, respectively. In HEK-293 and SK-N-SH cells, CEBPB significantly inhibited the relative fluorescence intensity. When we compared the relative fluorescence intensity of pGL3-HTR1A+pEGFP-N1-Basic with pGL3-HTR1A+pEGFP-N1-CEBPB, different sequence lengths of the *HTR1A* showed statistical significance. However, the degree of inhibition was different, indicating that CEBPB might specifically act on −1219 bp to −1209 bp. * 0.02 < *p* < 0.05, ** 0.00 < *p* <= 0.02, *** *p* <= 0.001.

**Figure 5 genes-10-00802-f005:**
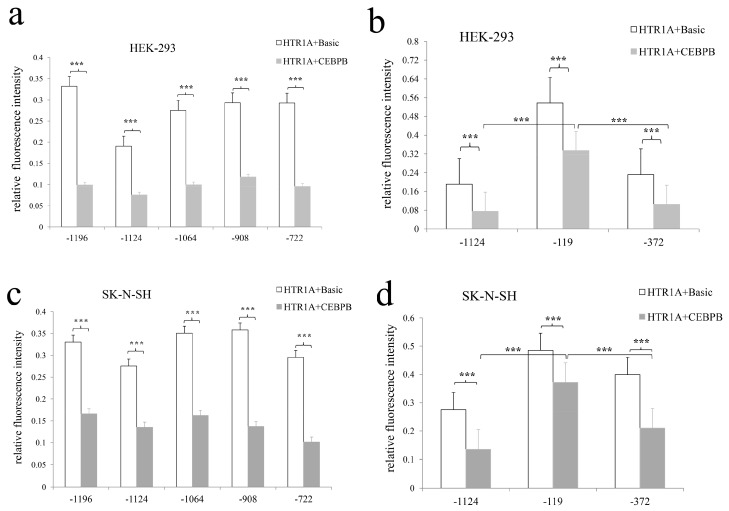
Effect of CEBPB on other fragments through the co-transfection of CEBPB with multiple target fragments in the two cell lines, we found that all sequences with the common region −722 bp–−372 bp strongly responded to CEBPB and were significantly inhibited by CEBPB. However, the inhibition degrees of the sequences were different. * 0.02 < *p* < 0.05, ** 0.00 < *p* <= 0.02, *** *p* <= 0.001.

**Figure 6 genes-10-00802-f006:**
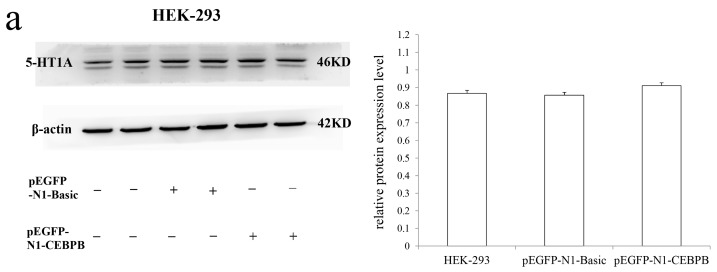
(**a**) Insignificant inhibitory effect of CEBPB on the endogenous 5-HT1A receptors expression, (**b**) Western blot were performed with three groups including the two cell lines protein, the protein which were extracted from the cell transfected with the basic vector or the over-expressing CEBPB vector. The gray values for each sample were normalized using β-actin (relative protein expression and showed as the mean ± SD. Student’s *t*-test was used for differential comparison. The results showed that CEBPB did not inhibit endogenous 5-HT1A receptor expression in both cell lines.

**Figure 7 genes-10-00802-f007:**
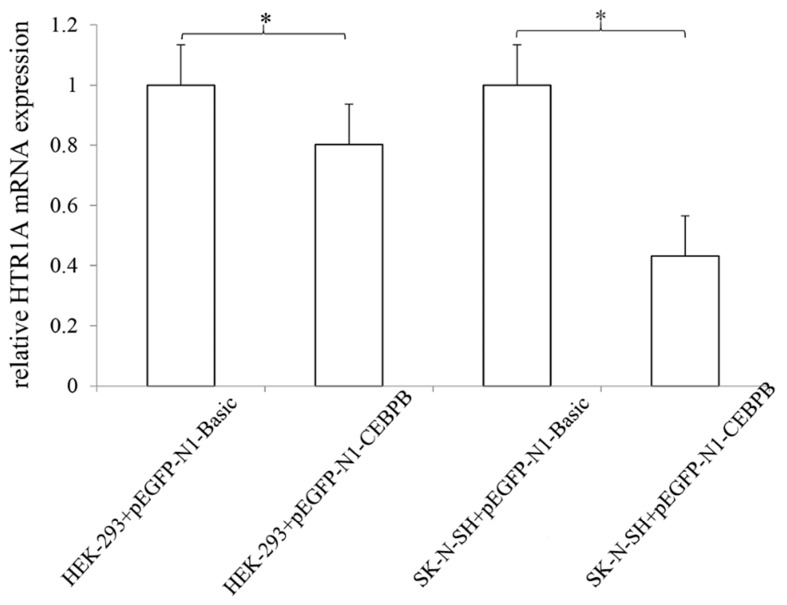
Inhibition effect of CEBPB on the 5-HT1A receptor mRNA expression. The effect of CEBPB on the expression of endogenous 5-HT1A receptor mRNA was detected by quantitative real-time PCR (qRT-PCR) and used the 2^−ΔΔCT^ method for statistics. The results showed that CEBPB significantly inhibited the mRNA expression in both cell lines. * 0.02 < *p* < 0.05.

**Table 1 genes-10-00802-t001:** Sequences of the probes in the electrophoretic mobility shift assay (EMSA).

Name	Sequences
*HTR1A*(−1235–−1196)	5′-ATTTCGTTCTCTCCCGGTTCCCCAACGTTAAAAAAAAAG-3′(WT)
mut2 (CEBPB)	5′-ATTTCGTTCTCTCCCAACCTTTTGGAGTTAAAAAAAAAG-3′(MT)
mut3 (FEV)	5′-CGGGATCCCTCTCCCGGTTCCCCAACGTTAAAAAAAAAG-3′(MT)
mut4(LMX1B)	5′-ATTTCGTTCTCTCCCGGTTCCCCAACGCCGGGCCGAAAG-3′(MT)

The sequences were the 5′-end biotin-labeled probes and the unlabeled specific, non-specific competitive probes synthesized in the EMSA experiment. The numbers indicated the positions of the probes in the *HTR1A*, and the underlines represented the bases of mutations. WT: wild type, MT: mutation type.

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
