# Peer review of "Transcription Factor CEBPB Inhibits the Expression of the Human HTR1A by Binding to 5′ Regulatory Region In Vitro"

_genes, 2019, doi:10.3390/genes10100802_

Round 1

Reviewer 1 Report

This paper presented interesting results that three 5’ regulatory regions of the HTR1A gene might negatively control the expression of HTR1A via being bound to transcription factor CCAA/T enhancer binding protein beta (CEBPB). This information could not only be added into the mechanism of regulation of HTR1A gene expression but is also of interest to pharmaceutical approaches to the treatment of HTR1A related diseases, especially neuropsychiatric disorders.

Several points make this work interesting. Firstly, these three targeted regions included two promoter regions (-1219 to -1209 bp and -722 to -372 bp) and one region that contains 99bp of exon (-119 to +99 bp). CEBPB potentially bound to three regions in the same gene, which might not be so common for other genes or other transcription factors. In addition, all recombinant vectors with different fragment lengths of HTR1A respond to CEBPB and relative fluorescence intensities reached statistical significance. Furthermore, the authors have carried out RT-qPCR and Western blot in order to study the inhibitory effect of CEBPB on endogenous 5-HT1A receptor expression. While protein expression of 5-HT1A receptors remained unchanged, mRNA expression of HTR1A decreased in these in vitro studies.

However, there are several questions that need to be further discussed or clarified in the paper:

Although the function of transcription factor (TF) CCAA/T enhancer binding protein beta (CEBPB) is not fully understood, in other studies, CEBPB showed positive regulatory effects in terms of gene expression. What is the potential mechanism that the 'enhancer/positive' TF showed a negative/inhibitory role on HTR1A expression in this study?

The effect of CEBPB on HTR1A expression experiments was not clearly illustrated in the paper or fully discussed. Why did all the transfected vectors respond to CEBPB if it only binds to certain regions? If the -1219 to -1209 bp region was the main binding region, why did transfected vectors without this region also show responses to CEBPB? Is it possibly due to the binding to the other two regions? If so, does this mean the other two regions have stronger effects than -1219 to -1209 bp region?

The results from Western blot were not consistent with RT-qPCR results. In the discussion, one possible explanation was that the sensitivity of Western blot (WB) is lower than the dual-luciferase reporter experiment. To support this argument, reference 33 was used. However, in the original paper, the sensitivity of WB antibody refers to the anti-CEBPB but not anti-HTR1A. In Figure 6, the 5-HT1A bands were moderately clear, thus the sensitivity of WB antibody might not be the issue.

The results showed in Figure 5 were not clear enough to let readers understand why -722 to -372 bp and -119 to +99 bp were the other two potential binding regions for CEBPB while all the other fragments were also significantly inhibited by CEBPB.

In their previous study (Wu, X., Xu, F. L., Ding, M., Zhang, J. J., Yao, J., & Wang, B. J. (2018). Characterization and functional analyses of the human HTR1A gene: 5’regulatory region modulates gene expression in vitro. BMC Genetics, 19(1), 115.), the -1235 - +99bp vs -1196 - +99bp fragments did not show significant expression effect, which indicated that -1235 - -1196bp region had no regulation effect for HTR1A expression. How do you explain that the -1219 - -1209bp in this study differs from that study?

Author Response

Reviewer 1,

Thank you very much for your careful review. We have uploaded the responses for your comments. Please see the attachment. Thanks again. Hope to get your support.

Reviewer 2 Report

In the current study, Xue Wu et al. tried to demonstrate the overexpression of CEBPB can inhibit human HTR1A gene expression. Actually, many arguments and suggestions are not clearly be addressed and described. Overall, in my opinion, the manuscript is not suitable and considerable to be accepted by Genes, at least in this version.

Major comments

Making and providing a distinguishable and reliable data are a scientist’s responsibility. For example, In Figure 2, especially in Fig. 2C, the authors should carefully redesign, redo and further describe the individual components in each shifted binding complex. Meanwhile, the sequences and differences of mutant DNA oligonucleotides should be represented side-by-side with wild-type DNA sequence. 3 is a relatively meaningless data. It is also not be described well for the issues including what is the rationale and the meaning of the result. In Fig. 4, in addition to using gain-of function assay to identify CEBPB resposive region, the CEBPB-mediated repressive effect on the promoter region of HTR1A gene should be identified by continuously shortening the so-called 5’-regulation region on HTR1A reporter until losing the CEBPB-mediated repressive effect. By the way, the standard transcriptional initiation site should be counted from transcription initiation site but not translational initiation site (ATG). The authors should deligently to figure out the transcription initiation site of HTR1A gene.

As well as experimental design and results, the writing and description in the manuscript has a large space to be improved.  

Author Response

Reviewer 2,

Thank you so much for your comments. We have modified the manuscript and uploaded it. We hope to get your support. Please see the attachment.

Reviewer 3 Report

In the submitted manuscript the Authors describe the identification of a transcription factor that binds to a regulatory region of the gene for the 5-HT1A receptor. Their results are potentially interesting for the understanding of the receptor expression.

With regard to the results described in the manuscript, they seem convincing and helpful to demonstrate the existence of the transcription factor, but at the same time the manuscript could benefit from a more schematic presentation of the results, a clearer description of the experiments and of their rationale, as well as a better organization of the figures showing the results of the experiments.

More specifically:

1) Although the manuscript is comprehensible in most parts, it could surely benefit from an English mother language revision: some sentences are unclear in their meaning.

2) In general, a different font size and spacing of figure legends with respect to those used in the main text could help in a better comprehension of the content of the manuscript;

3) The "legend" of Figure 5 seems more a part of the main text rather than a figure legend: in fact, it does not explain the content of the figure which it refers to.

4) The legend of Figure 6 shares the same problems of the legend of Figure 5. In addition the fonts and their sizes in this figure are not homogeneous. Furthermore some parts of the figure are not explained in the legend. Probably splitting the Figure in two parts (a+b separated from c) could increase its comprehension.

5) In the Discussion, lines 308-309, it is mentioned a result about the endogenous expression of 5-HT1A receptor at 24 up to 96 hours after transfection which it is not described in the Results section. Is it an original result of this paper? Why was it not mentioned as a "data not shown"?

In conclusion the manuscript could be accepted for publication provided that the above suggestions are included in the text.

Author Response

Reviewer 3,

Thank you for your serious review. We have corrected our manuscript according to your suggestions. We hope to get your support. Please see the attachment. Thanks again.
